# Germline *cis* variant determines epigenetic regulation of the anti-cancer drug metabolism gene dihydropyrimidine dehydrogenase (*DPYD*)

Ting Zhang[1], Alisa Ambrodji[2,3], Huixing Huang[1], Kelly J Bouchonville[1], Amy S Etheridge[4], Remington E Schmidt[1], Brianna M Bembenek[1], Zoey B Temesgen[1], Zhiquan Wang[5], Federico Innocenti[4], Deborah Stroka[6], Robert B Diasio[1], Carlo R Largiadèr[2], Steven M Offer[1,7,8]*†

[1]Department of Molecular Pharmacology and Experimental Therapeutics, Mayo Clinic, Rochester, United States; [2]Department of Clinical Chemistry, Inselspital, Bern University Hospital, University of Bern, Bern, Switzerland; [3]Graduate School for Cellular and Biomedical Sciences, University of Bern, Bern, Switzerland; [4]Eshelman School of Pharmacy, Division of Pharmacotherapy and Experimental Therapeutics, University of North Carolina at Chapel Hill, Chapel Hill, United States; [5]Division of Hematology, Department of Medicine, Mayo Clinic, Rochester, United States; [6]Department of Visceral Surgery and Medicine, Inselspital, Bern University Hospital, University of Bern, Bern, Switzerland; [7]Department of Pathology, University of Iowa Carver College of Medicine, University of Iowa, Iowa City, United States; [8]Holden Comprehensive Cancer Center, University of Iowa Carver College of Medicine, University of Iowa, Iowa City, United States

*For correspondence:
soffer@uiowa.edu

†Lead contact

**Abstract** Enhancers are critical for regulating tissue-specific gene expression, and genetic variants within enhancer regions have been suggested to contribute to various cancer-related processes, including therapeutic resistance. However, the precise mechanisms remain elusive. Using a well-defined drug-gene pair, we identified an enhancer region for dihydropyrimidine dehydrogenase (DPD, *DPYD* gene) expression that is relevant to the metabolism of the anti-cancer drug 5-fluorouracil (5-FU). Using reporter systems, CRISPR genome-edited cell models, and human liver specimens, we demonstrated in vitro and *vivo* that genotype status for the common germline variant (rs4294451; 27% global minor allele frequency) located within this novel enhancer controls *DPYD* transcription and alters resistance to 5-FU. The variant genotype increases recruitment of the transcription factor CEBPB to the enhancer and alters the level of direct interactions between the enhancer and *DPYD* promoter. Our data provide insight into the regulatory mechanisms controlling sensitivity and resistance to 5-FU.

## eLife assessment

This manuscript presents **valuable** findings on the identification of epigenetically mediated control for the recognition of dihydropyrimidine dehydrogenase (DPYD) gene expression that is linked with cancer treatment resistance using 5-fluorouracil. The evidence is **compelling**, supported by data from patient-derived specimens and direct assessment of 5-fluorouracil sensitivity, which provides confidence in the proposed mechanisms. The model is additionally supported by genome data from

a population with high "compromised allele frequency". This work will interest those studying drug resistance in cancer therapy.

## Introduction

Therapeutic resistance has been reported for nearly all anti-cancer drugs (*Ramos et al., 2021*), and as many as 90% of mortalities in cancer can be linked to drug resistance (*Mansoori et al., 2017*). Enhancer-mediated regulation of gene expression has been increasingly implicated in multiple cancer-related processes, including resistance, response, and toxicity to cancer therapies (*Koutsi et al., 2022*; *Lauschke et al., 2019*). However, the specific molecular mechanisms through which epigenetic changes drive these processes remain mostly elusive.

Enhancers regulate gene expression by recruiting transcription factors (TFs) and subsequently interacting with the promoter region of a target gene to drive expression. The activity of enhancers can be affected by localized epigenetic state and genetic variations that affect TF binding. More than 90% of disease-associated genetic variants identified in genome-wide association studies (GWAS) lie in the non-coding portion of the genome (*Manolio et al., 2009*; *Maurano et al., 2012*). Many of these GWAS variants map to putative enhancers (*Boix et al., 2021*). Regardless, translating enhancer GWAS variants into disease mechanisms remains a largely unmet challenge (*Claringbould and Zaugg, 2021*).

In previous studies, we identified a *cis* expression quantitative trait locus (eQTL) for the chemotherapeutic metabolism gene dihydropyrimidine dehydrogenase enzyme (DPD, encoded by the *DPYD* gene) (*Etheridge et al., 2020*). DPD is the initial and rate-defining enzyme for the conversion of the commonly used chemotherapeutic 5-FU into inactive metabolites. Hepatic DPD eliminates approximately 85% of circulating 5-FU within minutes of administration (*Sommadossi et al., 1982*; *Heggie et al., 1987*), and deficiency of DPD is associated with severe (clinical grade ≥3) toxicity to 5-FU (*Amstutz et al., 2018*). Clinical and preclinical studies have identified missense and splicing variants of *DPYD* that consistently correlate with reduced DPD function and increased 5-FU toxicity risk (*Amstutz et al., 2018*; *Lee et al., 2014*; *Shrestha et al., 2018*; *Offer et al., 2014b*; *Offer et al., 2013a*; *Offer et al., 2013b*; *Nie et al., 2017*). However, these variants explain <20% of reported cases of grade ≥3 5 FU toxicities (*Amstutz et al., 2018*; *Meulendijks et al., 2015*), and the mechanisms contributing to 5-FU resistance, and therefore means to overcome resistance, remain elusive.

In the present study, we used human liver tissues and cellular models to identify and characterize a novel *cis*-enhancer element capable of modulating *DPYD* expression. We additionally provide mechanistic data demonstrating that transcription-factor driven expression of *DPYD* is dependent on allele status for a common germline genetic variant located within this enhancer, suggesting that the variant could be a valuable biomarker of 5-FU toxicity risk. Furthermore, the genotype-dependent regulation of DPD expression offers a potential mechanism for overcoming resistance in patients carrying the risk variant.

## Results

### Identification of *DPYD* enhancer regions

Candidate *cis* enhancer regions for *DPYD* (NM_000110.4) were identified using GeneHancer (*Fishilevich et al., 2017*) and Ensembl Regulatory Build (*Zerbino et al., 2015*) data mapped to GRCh37/hg19. We defined the upstream boundary of the target region using enhancer data supported by multiple methods and gene-enhancer links supported by multiple methods as implemented within GeneHancer (i.e. putative enhancer regions with 'double-elite' status; *Figure 1A*). We then mapped annotated transcription factor binding sites in the target region using ChIP-seq data from the Ensembl Regulatory Build as an additional layer of evidence (*Figure 1A*). In all, three candidate proximal upstream enhancer regions were identified, which will be referred to herein as E9, E16, and E20 based on the relevant location upstream of the *DPYD* transcription start site (*Figure 1A*).

CRISPR-interference (CRISPRi) and -activation (CRISPRa) were used to determine which of these regions were capable of regulating *DPYD* expression. For CRISPRi, we generated cell lines that stably expressed the Krüppel-associated box (KRAB) domain of Kox1 fused to nuclease-deficient Cas9 protein (dCas9-KRAB; *Yeo et al., 2018*). For CRISPRa, we created cell lines that stably expressed dCas9 fused to the histone acetyltransferase E1A binding protein P300 (dCas9-P300; *Hilton et al.,*

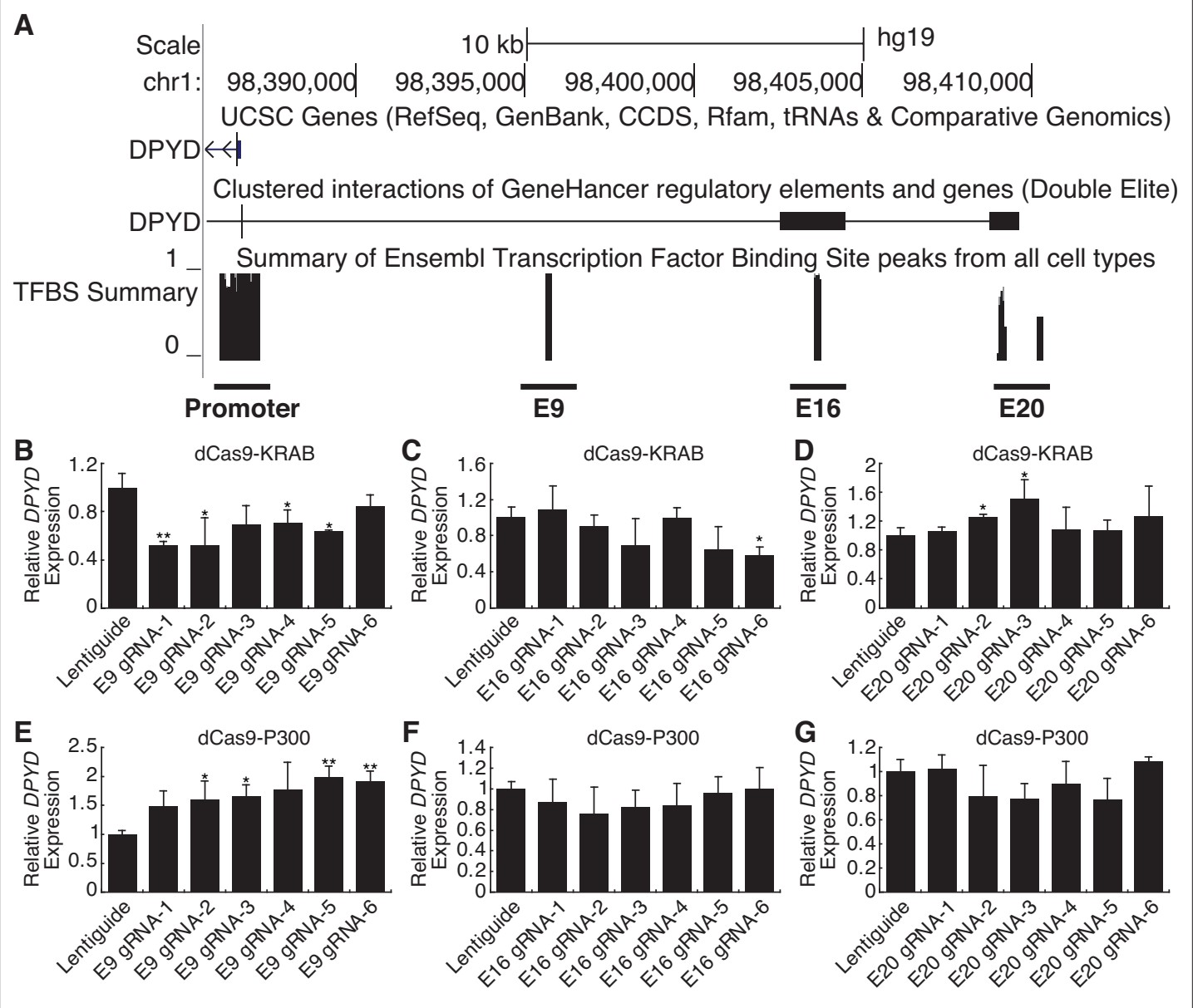

**Figure 1.** Identification of a novel dihydropyrimidine dehydrogenase (*DPYD*) enhancer. (**A**) Candidate *DPYD* enhancer regions were selected for further study using data from GeneHancer and Ensembl Regulatory Build. Coordinates are based on GRCh37/hg19. Regions are termed E9, E16, and E20 based on the approximate distance upstream of the *DPYD* transcription start site. For CRISPR inhibition (CRISPRi), *DPYD* expression was measured in HepG2 cells expressing dCas9-KRAB following transfection with guide-RNAs specific to the E9 (**B**), E16 (**C**), and E20 (**D**) regions. For CRISPR activation (CRISPRa), *DPYD* expression was measured in HepG2 cells expressing dCas9-P300 following transfection with guide-RNAs specific to E9 (**E**), E16 (**F**), and E20 (**G**). Data represent the mean of three independent biological replicates ± SD. *p<0.05; **p<0.005. p-values were calculated using a two-tailed Student's t-test comparing results to those from lentiguide controls.

The online version of this article includes the following figure supplement(s) for figure 1:

**Figure supplement 1.** CRISPR interference (CRISPRi) and CRISPR activation (CRISPRa) screen to identify dihydropyrimidine dehydrogenase (*DPYD*) *cis*-regulatory elements in HCT116 cells.

*2015*). Six guide RNAs (gRNAs) were used to target dCas9 fusion proteins to each region of interest, and quantitative reverse transcription PCR (qRT-PCR) was used to monitor changes in *DPYD* expression following transfection.

Targeting of dCas9-KRAB (CRISPRi) to the E9 region significantly decreased *DPYD* expression relative to control for four of six gRNAs (*Figure 1B*). Targeting of dCas9-KRAB to E16 resulted in

a significant decrease in *DPYD* expression for one of six gRNAs (*Figure 1C*), and no significant reductions in *DPYD* expression were noted following transfection of gRNAs for E20 in dCas9-expressing cells; however, two gRNAs increased *DPYD* expression (*Figure 1D*). For reciprocal CRISPRa experiments, targeting dCas9-P300 to E9 significantly increased *DPYD* expression for four of six gRNAs tested (*Figure 1E*), whereas gRNAs specific to E16 and E20 did not elicit any significant changes in *DPYD* expression (*Figure 1F and G*). Consistent with the results in HepG2 cells, CRISPRi and CRISPRa targeted to the E9 region significantly altered *DPYD* expression in HCT116 cells (*Figure 1—figure supplement 1A and D*), whereas no significant changes were noted when targeted to E16 or E20 (*Figure 1—figure supplement 1B, C, E, F*).

To confirm that the expected epigenetic changes were induced at the E9 region, we performed chromatin immunoprecipitation (ChIP) coupled with quantitative PCR (ChIP-qPCR) in dCas9-KRAB and -P300 expressing cells transfected with E9 gRNAs. As expected, targeting dCas9-KRAB to E9 resulted in a localized increase in H3K9me3 compared to controls, indicating a shift from active to inactive chromatin at E9 (*Figure 2A*; the primer nearest E9 is denoted by the gray box). Targeting of dCas-P300 to E9 caused changes consistent with epigenetic activation of the region, including a localized gain of H3K27ac-marked chromatin (*Figure 2B*). Similar changes in localized histone modifications were noted when dCas9-KRAB and dCas9-P300 HCT116 cells were directed to E9 (*Figure 2C–D*). Collectively, these results indicate that the E9 region might act as a *cis*-regulatory element modulating *DPYD* expression.

## Identification of rs4294451 as a putative regulatory SNP within E9

In an expression quantitative trait loci (eQTL) genome-wide association study (GWAS) conducted in human liver specimens, Etheridge et al. previously identified three independent haplotype blocks that significantly associated with altered *DPYD* expression (*Etheridge et al., 2020*). One of the blocks identified in that study spanned the E9 region, prompting us to investigate if genetic variants in E9 could potentially perturb *DPYD* regulation. Based on the biological role of enhancers, we postulated that a causal functional single nucleotide polymorphism (SNP) could interfere with transcription factor binding. Within the Ensembl Regulatory Build (*Zerbino*

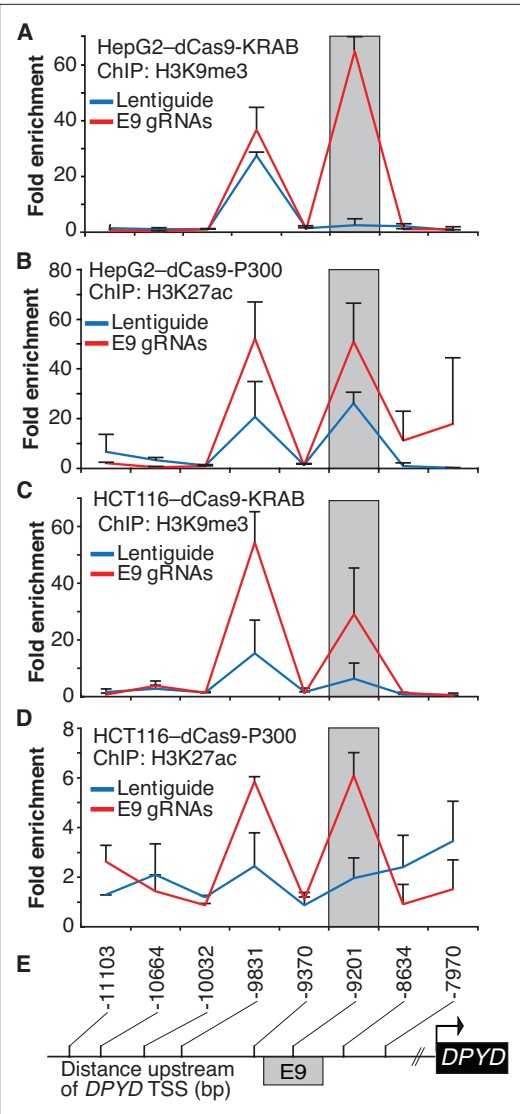

**Figure 2.** Epigenetic changes at the E9 region induced by CRISPR interference (CRISPRi)/CRISPR activation (CRISPRa). Lentiguide vectors encoding guide-RNAs targeting the E9 region (E9 gRNAs) or empty vector control (Lentiguide) were transduced into HepG2 cells expressing dCas9-KRAB (**A**) or dcas9-P300 (**B**) and HCT116 cells expressing dCas9-KRAB (**C**) or dcas9-P300 (**D**). Chromatin immunoprecipitation (ChIP) was performed using antibodies specific to H3K9me3 (**A, C**) or H3K27ac (**B, D**). Quantitative PCR using primers centered at the indicated regions (**E**) was used to measure the relative abundance of H3K9me3 and H3K27ac. Data are presented relative to input DNA control and are further normalized to IgG control. Error bars represent the SD of three independent biological replicate experiments.

The online version of this article includes the following figure supplement(s) for figure 2:

**Figure supplement 1.** Genomic context of rs4294451.

*et al., 2015*), we identified a region of approximately 150 bp showing evidence for binding by multiple transcription factors in HepG2 cells, but not in other cell types tested (*Figure 2—figure supplement 1*). Within this region, we identified the variant rs4294451, which is a tag SNP for the previously identified eQTL block (*Etheridge et al., 2020*), prompting us to hypothesize that this variant might be the causal eQTL SNP.

To assess the contribution of rs4294451 alleles to *DPYD* regulation, we first performed reporter assays by cloning the *DPYD* promoter and E9 enhancer region containing either the reference or variant allele for rs4294451 (T or A, respectively) into a luciferase reporter vector (*Figure 3A*). Reporter vectors containing the T-allele yielded significantly higher luciferase activity compared to vectors containing only the *DPYD* promoter (p=0.00054), whereas those containing the A allele within E9 showed a more modest increase (p=0.070, *Figure 3B*). Directly comparing E9-containing reporters, vectors containing the T allele showed significantly higher activity than those with the A allele (p=0.046, *Figure 3B*), suggesting that the variant could impact *DPYD* expression.

## Impact of rs4294451 genotype on *DPYD* expression and regulation

To directly characterize the impact of rs4294451 alleles on *DPYD* expression at the endogenous locus, we used CRISPR-mediated genome editing to create matched isogenic HCT116 knock-in cell models for each genotype. Consistent with the results from the reporter assay (*Figure 3B*), cells homozygous for the A allele had significantly lower *DPYD* expression compared to cells that were homozygous for the T allele ($P$=0.0011, *Figure 3C*). Heterozygous rs4294451 A/T cells displayed intermediate expression compared to cells with homozygous A/A and T/T genotypes (p=0.00013 and p=0.028, respectively; *Figure 3C*). At the chromatin level, cells homozygous for the A allele showed reductions in H3K27ac and accumulation of H3K9me at E9 compared to cells that were homozygous or heterozygous for the T allele (*Figure 3D–E*). This finding is consistent with the A allele being associated with a less active epigenetic state at E9. Cells carrying a heterozygous genotype for rs4294451 displayed an intermediate epigenetic state, which is consistent with one epigenetically active and one inactive copy of E9 (*Figure 3D–E*). Similar epigenetic differences between rs4294451 genotypes were noted at E9 in liver specimens obtained from human donors (*Figure 3—figure supplement 1*).

## Allele-specific expression of *DPYD* transcripts

To precisely determine the impact of rs4294451 alleles on *DPYD* expression in vivo, we measured allele-specific expression using reverse transcriptase digital droplet PCR (RT-ddPCR) using donor liver tissues. Because rs4294451 falls outside of the coding region, we used coding region variants as proxies to measure strand-specific expression. These coding variants consisted of *DPYD*-c.85T>C (rs1801265) and c.496A>G (rs2297595), both of which are in linkage disequilibrium (LD) with the rs4294451 T allele (c.85C: D'=0.92, $R^2$=0.80; c.496G: D'=0.67, $R^2$=0.21). Imperfect LD between rs4294451 and proxies allowed us to use samples that are heterozygous for the coding region SNP, but homozygous for rs429441 genotype, as an additional level of control. In samples measured using the c.85 genotype, the fractional abundance of the C-allele ranged from 0.542 to 0.602 in samples heterozygous for rs4294451 (*Figure 3F*). Expressed in terms of relative expression, the C-allele was expressed at levels 18–51% higher than the T-allele (i.e. 0.542/0.458=1.18 and 0.602/0.398=1.51). In contrast, the sample that was homozygous for the rs4294451 T allele showed allelic expression at the c.85 locus that was indistinguishable from the DNA control (*Figure 3F*), indicating equal expression of both transcripts. In measurements using c.496, a significant allelic-imbalance of 0.565 (CI: 0.546–0.584) in favor of the variant c.496 G-allele compared to the DNA control (CI: 0.488–0.511) was observed (*Figure 3G*). This again indicated higher expression of the linked rs4294451 T allele. Notably, liver specimens that were homozygous at the rs4294451 locus again showed c.496 allelic expression that was indistinguishable from the DNA control (*Figure 3G*). These data indicate that rs4294451 significantly affects the expression of *DPYD* in the liver.

## Rs4294451 genotype affects interactions between the *DPYD* promoter and E9

We next investigated the impact of rs4294451 genotype on intra-chromatin interactions between the E9 region and the *DPYD* promoter using quantitative analysis of chromatin conformation capture (3C-qPCR) in knock-in cells. The schematic in *Figure 4A* shows the location of primers used for 3C-qPCR

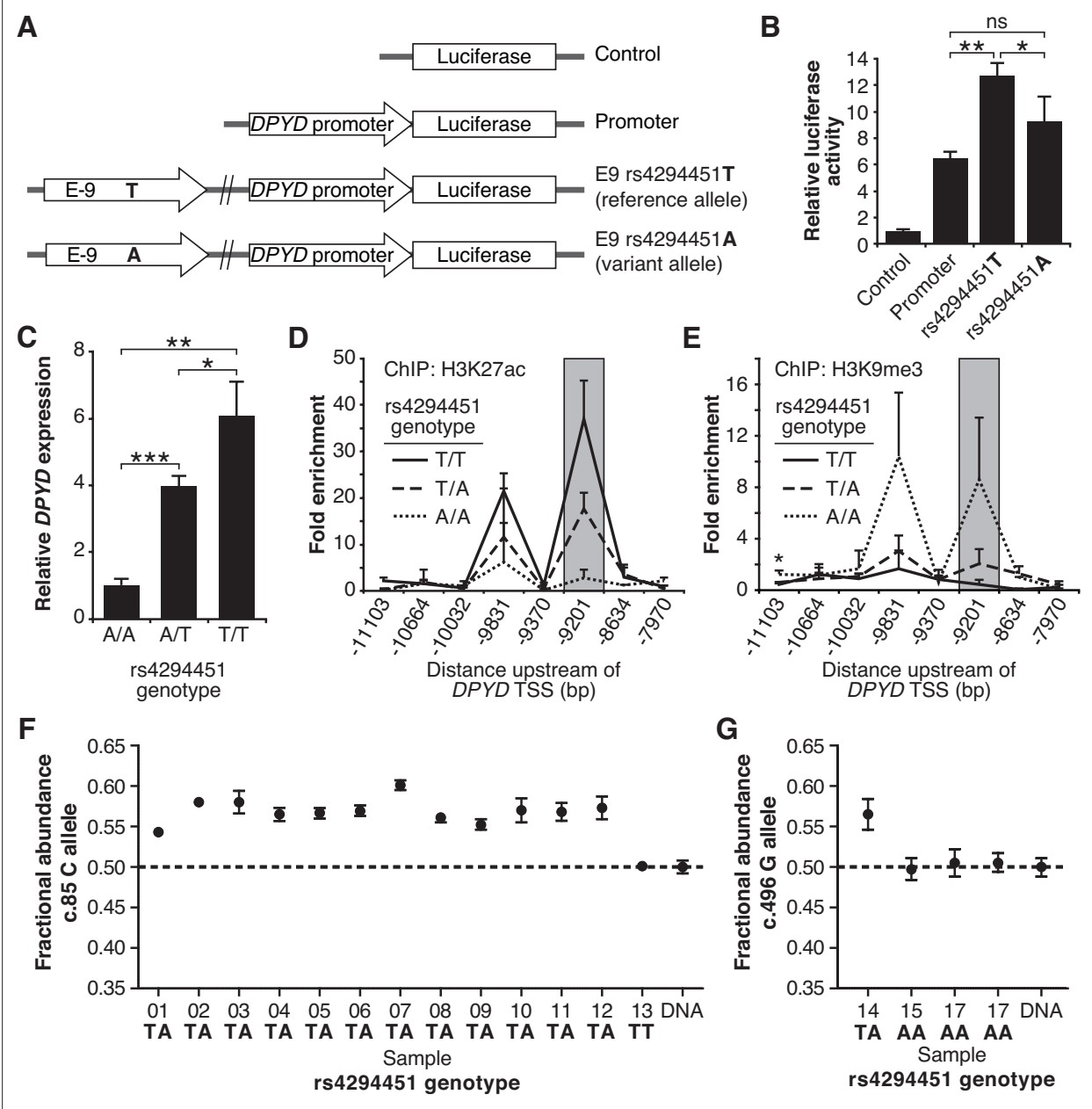

**Figure 3.** The rs4294451 T allele is associated with elevated dihydropyrimidine dehydrogenase (*DPYD*) expression. (**A**) Luciferase reporter constructs were generated by cloning the E9 region containing the reference rs4294451 T allele or the variant rs4294451 A allele into reporter vectors containing the *DPYD* promoter. (**B**) Luciferase reporter activity was measured for vectors shown in panel A. Error bars represent the SD of three independent biological replicates. (**C**) Relative expression of endogenous *DPYD* measured via RT-qPCR in HCT116 cells engineered using CRISPR-mediated genome editing to contain the depicted genotypes for rs4294451. Chromatin enrichment of H3K27ac (**D**) and H3K9me3 (**E**) was measured using chromatin immunoprecipitation (ChIP) coupled with quantitative PCR (ChIP-qPCR) in HCT116 cells engineered to contain the indicated genotypes at rs4294451. (**F**) *DPYD* allele-specific expression was measured in human liver tissues using the C and T alleles at position c.85. (**G**) Allele-specific expression was measured using the c.496-A and G alleles. All panels: *p<0.05; **p<0.005; ***p<0.0005. p-values were calculated as pairwise comparisons between the indicated groups using a two-tailed Student's t-tests. Error bars represent SD.

The online version of this article includes the following figure supplement(s) for figure 3:

**Figure supplement 1.** Rs4294451 A allele is associated with epigenetic repression at the E9 region in human liver specimens.

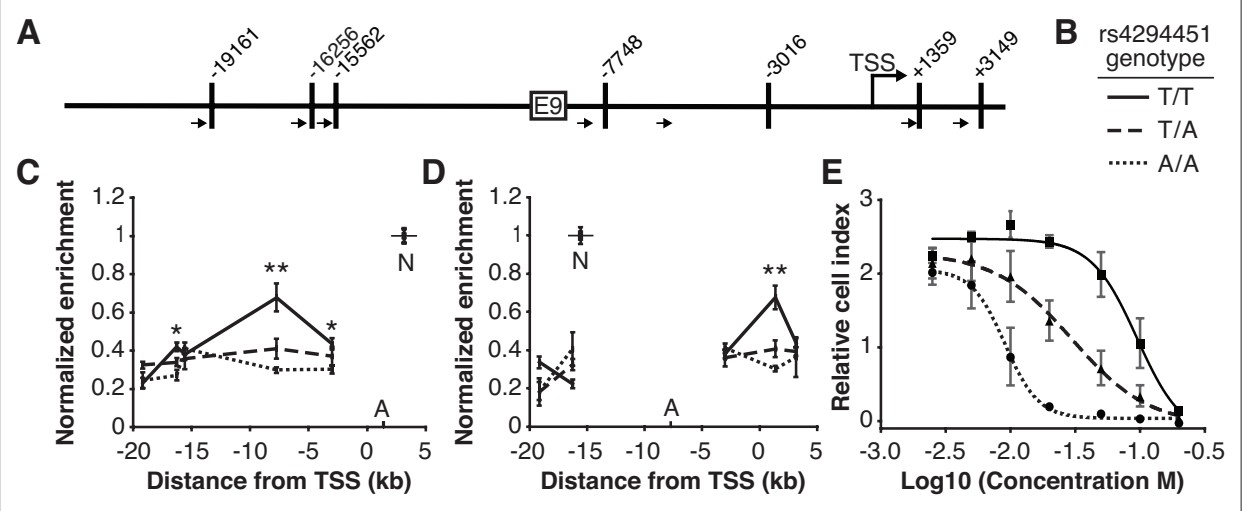

**Figure 4.** The rs4294451 T allele is associated with increased interaction between E9 and the dihydropyrimidine dehydrogenase (*DPYD*) promoter. (**A**) Schematic of *Hind*III restriction enzyme sites (vertical bars) and primers (arrows) used for chromatin conformation capture (3 C) relative to the *DPYD* transcription start site (TSS) and E9 region. (**B**) Legend for panels C–E. (**C**) 3 C of chromatin interactions in rs4294451 knock-in HCT116 cells using anchor primer positioned within the digestion fragment containing the *DPYD* promoter. A, location of anchor primer; N, location of primer used for data normalization. (**D**) 3 C of knock-in cells using anchor primer positioned within the fragment containing the E9 region. For panels C–D: one-way ANOVA p: *p<0.01; **p<0.001; all other data points, p>0.01. (**E**) Knock-in cells were treated with dilutions of 5-fluorouracil (5-FU) and viability was assessed using real-time cell analysis (RTCA). Cell index is a measure of impedance between electrodes that are arrayed at the bottom of the RTCA plate and is representative of the number of live cells attached to the culture plate. Data are from 48 hr of 5-FU treatment. For all plotted data, the mean ± SD of three independent replicates is presented.

relative to the *DPYD* transcription start site (TSS) and E9 region, as well as the location of *Hind*III digestion sites. When using primers anchored at the *DPYD* promoter, the E9 region showed a significantly stronger interaction with the promoter in cells carrying the T allele compared to those homozygous for the A allele (p=8.9 × 10$^{-4}$ comparing TT to AA; ANOVA p=3.0 × 10$^{-4}$ across all three genotypes; *Figure 4C*). With primers anchored to the E9 region, cells homozygous for the rs4294451 T allele showed stronger promoter interaction than those carrying the A allele (p=4.9 × 10$^{-4}$ comparing TT to AA; ANOVA p=1.2 × 10$^{-4}$ across all three genotypes; *Figure 4D*).

### Rs4294451 genotype confers differential sensitivity to 5-FU

Having demonstrated that the rs4294451 genotype impacts *DPYD* expression, we next evaluated the impact on 5-FU sensitivity using real-time cellular analysis (RTCA). RTCA provides a measure of the number of viable cells present in a culture. We have previously demonstrated the utility of this technology for measuring differences in drug-sensitivity to 5-FU and that those differences directly correlate with DPD enzyme function within cells (*Shrestha et al., 2018*; *Offer et al., 2013b*). Consistent with lower expression of *DPYD*, knock-in cells homozygous for the rs4294451 A allele were significantly more sensitive to 5-FU than cells homozygous for the T allele (IC$_{50}$ concentrations were 9.1 and 95.0 µM 5-FU, respectively; p<0.0001; *Figure 4E*). Heterozygous T/A cells showed an intermediate IC$_{50}$ value of 29.9 µM 5-FU (*Figure 4E*).

### Transcription factor binding within E9 is affected by rs4294451 status

Based on ENCODE data, rs4294451 is localized to a region that previously has been shown to be bound by multiple transcription factors (*Figure 2—figure supplement 1*). To determine if known transcription factor binding sites could be affected by allele status at rs4294451, we computed binding scores for JUND, MYBL2, HNF4A, CEBPB, FOXA1, and FOXA2 for sequences comprising the E9 region containing the T and A alleles at rs4294451. CEBPB, FOXA1, and FOXA2 showed differential predicted binding scores between the rs4294451 A- and T-containing query sequences (data not shown). For CEBPB, the differential binding site had a higher score than other sites in the queried region that did not overlap with rs4294451, suggesting that the SNP could affect CEBPB binding in the region. While predicted binding in the region varied for FOXA1 and FOXA2 depending on

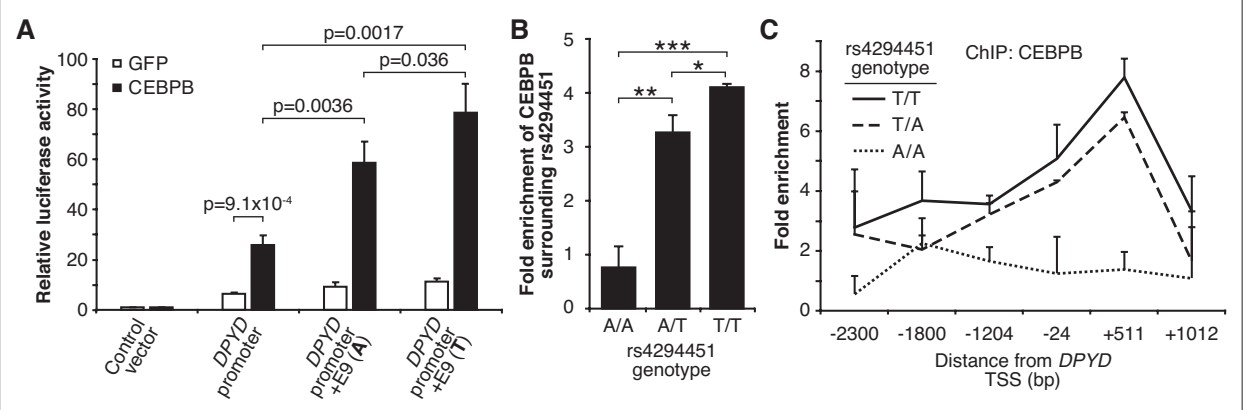

**Figure 5.** T allele is associated with higher occupancy of CEBPB at E9 and the dihydropyrimidine dehydrogenase (*DPYD*) promoter. (**A**) Expression plasmids for CEBPB or GFP (control) were co-transfected into HEK293T cells with the luciferase reporter plasmids depicted in *Figure 3A*. (**B**) Chromatin immunoprecipitation (ChIP) coupled with quantitative PCR (ChIP-qPCR) was performed to examine the relative CEBPB enrichment at the DNA fragments within the E9 region in knock-in HCT116 cells for the rs4294451 genotypes indicated. Primers used for E9 are centered on the position at 9201 nucleotides upstream of the *DPYD* TSS. (**C**) ChIP-qPCR was used to measure CEBPB occupancy surrounding the *DPYD* promoter region. *p<0.05; **p<0.005; ***p<0.0005. p-values were calculated as pairwise comparisons between the indicated groups using a two-tailed Student's t-tests. For all panels, error bars represent the SD of three independent replicates.

rs4294451 genotype, other binding sites that did not overlap with the variant site showed stronger predicted binding, indicating that the SNP was likely not affecting the critical binding site in the region. Binding scores did not differ by genotype for JUND or HNF4A, and no binding site above the threshold score was detected in the region for MYBL2.

Based on the above, we sought to determine if CEBPB could regulate *DPYD* expression through the E9 region and if regulation was affected by rs4294451 genotype. We first used reporter assays in conjunction with CEBPB. CEBPB expression significantly increased reporter activity relative to GFP control for luciferase vectors containing the *DPYD* promoter, indicating that the promoter contains CEBPB recognition sites (p=9.1 × 10⁻⁴; *Figure 5A*). In cells overexpressing CEBPB, plasmids containing the promoter and E9 region resulted in higher luciferase activity compared to those containing only the *DPYD* promoter, regardless of the presence of the A or T allele (p=0.0036 and p=0.0017, respectively), suggesting that rs4294451 genotype does not completely disrupt CEBPB regulation through E9 (*Figure 5A*). Comparing results for the A and T alleles at rs4294451 in the presence of overexpressed CEBPB indicates that the T allele is likely more responsive to CEBPB (*Figure 5A*; p=0.036).

To directly examine CEBPB binding to the E9 region and to determine if binding affinity differs between the rs4294451 A and T alleles, we performed ChIP with CEBPB antibodies in isogenic knock-in HCT116 cells for the rs4294451 A/A, A/T, and T/T genotypes. Cells homozygous for the reference T allele showed significantly higher CEBPB occupancy at both E9 and the *DPYD* promoter than A/T or A/A cells (*Figure 5B–C*). These results suggest that rs4294451 genotype determined the binding potential for CEBPB at the E9 enhancer region, which, in turn, affects CEBPB-driven expression of *DPYD* from the promoter.

## Upregulation of *DPYD* expression by CEBPB is dependent on the rs4294451-T allele

To further characterize the role of CEBPB in regulating *DPYD*, we disrupted CEBPB expression using two independent shRNAs (denoted as 'sh1' and 'sh2') in HCT116 knock-in cells carrying rs4294451 A/A, A/T, and T/T genotypes. Knockdown of CEBPB was confirmed at the protein (*Figure 6A*) and mRNA (*Figure 6B–D*) levels. Knockdown of CEBPB significantly reduced *DPYD* expression in rs4294451 T/A and T/T cells, but not in rs4294451 A/A cells (*Figure 6E–F*). CEBPB knockdown also reduced occupancy of the transcription factor at both E9 and the *DPYD* TSS in rs4294451 A/T and T/T cells, but not A/A cells, and the level of CEBPB occupancy at both regions under CEBPB knockdown conditions is similar to that in control shRNA-treated A/A cells (*Figure 6—figure supplement 1*).

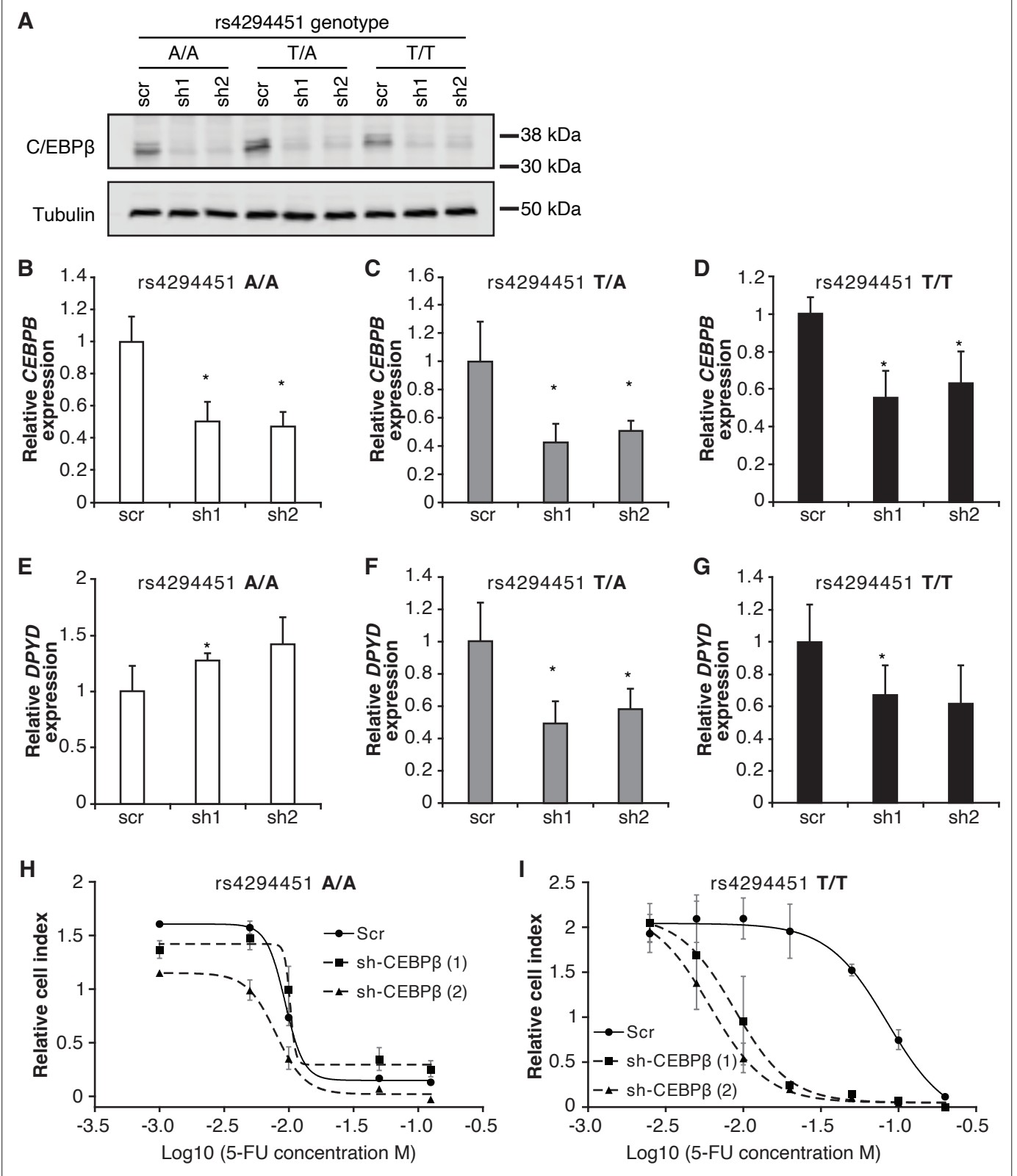

**Figure 6.** CEBPB-mediated upregulation of dihydropyrimidine dehydrogenase (*DPYD*) is dependent on rs4294451 T allele. (**A**) Immunoblot showing knockdown of CEBPB expression in HCT116 cells carrying different rs4294451 genotypes transduced with lentiviral particles encoding two independent shRNAs against CEBPB (sh1 and sh2) or a scrambled control shRNA (scr) (see also *Figure 6—source data 1*, *Figure 6—source data 2*, and *Figure 6—source data 3*). *CEBPB* expression was measured by RT-qPCR in HCT116 A/A (**B**), T/A (**C**) and T/T cells (**D**) transduced with the indicated shRNA

*Figure 6 continued on next page*

*Figure 6 continued*

lentiviral particles. *DPYD* expression was measured in shCEBPB and scramble control HCT116 A/A (**E**), T/A (**F**), and T/T (**G**) cells. The effect of CEBPB knockdown on cell viability in HCT116 A/A cells (**H**) and HCT116 T/T cells (**I**) was measured by RTCA. Data shown are from 48 hr of 5-fluorouracil (5-FU) treatment at the indicated concentrations. *p<0.05, calculated as a pairwise two-sided Student's t-test comparing the indicated data to that of the associated scr control. Error bars represent the SD from three independent replicates.

The online version of this article includes the following source data and figure supplement(s) for figure 6:

**Source data 1.** Original file for western blot analysis in *Figure 6A* (anti-CEBPB).

**Source data 2.** Original file for western blot analysis in *Figure 6A* (anti-tubulin).

**Source data 3.** PDF containing *Figure 6A* and annotated western blots used to make figure, including highlighted bands and sample labels.

**Figure supplement 1.** Disruption of CEBPB reduces enrichment at the E9 region and dihydropyrimidine dehydrogenase (*DPYD)* promoter in rs4294451 T/T and A/T cells, but not in T/T cells.

**Figure supplement 2.** Allele frequency for rs4294451-T allele in global populations.

We next determined the extent to which CEBPB contributed to 5-FU chemoresistance in cells with the T allele of rs4294451 (*Figure 4E*). In rs4294451 A/A cells, knockdown of CEBPB did not affect 5-FU sensitivity (*Figure 6H*). In contrast, CEBPB knockdown significantly reduced the $IC_{50}$ for 5-FU in rs4294451 T/T cells (p<0.0001; *Figure 6I*).

## The rs4294451-T allele is enriched in individuals of African ancestry

Data from the Genome Aggregation Database (gnomAD *Chen et al., 2022*) were used to estimate the frequency of the rs4294451-T allele in global populations (*Figure 6—figure supplement 2A*). The highest minor allele frequency (MAF) was noted for African/African American individuals (40% MAF), where the MAF was lowest in East Asian individuals (7%). For comparison, individuals of European (Non-Finnish) ancestry, the population with the highest number of individuals reported in gnomAD, had a MAF of 23%. Within Latino-Admixed American gnomAD subjects, similar differences in MAFs were noted in local ancestry-informed frequency data (*Figure 6—figure supplement 2B*).

## Discussion

The antitumor efficacy and risk of severe adverse events associated with 5-FU are determined by the overall bioavailability of the drug in plasma. As the critical determinant of 5-FU pharmacokinetics, liver DPD expression is pivotal to both the risk of severe adverse events and therapeutic resistance in 5-FU chemotherapy at opposite ends of the exposure spectrum. This is underscored by the narrow therapeutic window for 5-FU, with toxicity and efficacy occurring at partially overlapping drug exposure levels (*Beumer et al., 2019*). Deleterious germline coding-region *DPYD* variants have been linked to severe 5-FU toxicity *Amstutz et al., 2018*; *Lee et al., 2014*; however, these variants are responsible for only a small fraction of severe adverse events in 5-FU use and are unlikely to contribute to drug resistance (*Henricks et al., 2018*). Elevated expression of DPD in tumor cells is known to confer 5-FU resistance (*Jiang et al., 1997*; *Kikuchi et al., 2015*), and upregulation of hepatic DPD expression has been shown to reduce drug exposures and promote the development of 5-FU–resistant tumors (*Li et al., 2013*). However, the mechanisms regulating DPD expression are not well characterized, nor is it understood how the regulatory processes can be altered to support the development of 5-FU resistance.

In the current study, we identified a novel *cis*-enhancer region for *DPYD* that is located approximately 9 kb upstream from the gene's transcription start site. We additionally provide evidence that the E9 region directly interacts with the *DPYD* promoter, supporting E9 as a functional enhancer for *DPYD* expression (*Figure 4*). We demonstrated that CEBPB is a critical transcription factor for *DPYD* that binds to this enhancer region, termed E9, promoting enhancer-promoter interactions and increasing *DPYD* expression. We also showed that the allele status of the germline variant rs4294451, located within the E9 region, can affect CEBPB-driven *DPYD* expression and sensitivity/resistance to 5-FU, making it a strong candidate biomarker for 5-FU toxicity risk and potentially tumor resistance to 5-FU–based cancer therapy. These findings are consistent with the recent identification of a haplotype block linked to the rs4294451 T allele that was significantly associated with elevated *DPYD* expression in human liver tissues (*Etheridge et al., 2020*). In the present manuscript, we show that the rs4294451

T allele is enriched for active chromatin marks in both human liver (*Figure 3—figure supplement 1*) and in cellular knock-in models (*Figure 3D–E*). Furthermore, using an innovative digital droplet RT-qPCR–based approach, we demonstrate that the rs4294451 T allele is associated with elevated expression of the *cis DPYD* transcript in human liver (*Figure 3F–G*).

Allele frequency data retrieved from gnomAD suggest that a majority (65–70%, estimated from allele frequencies) of individuals of African ancestry carry the rs4294451-T allele (*Figure 6—figure supplement 2*), whereas only about 35% of individuals of European ancestry are predicted to be carriers of the T allele. African American patients have worse overall survival in colorectal cancer compared to white patients, owing to biological and non-biological factors. While differential access to healthcare, treatment bias, and socioeconomic factors have been shown to contribute to the poorer prognosis (*Mayberry et al., 1995*), other unrecognized factors also contribute to this difference (*Govindarajan et al., 2003*; *Alexander et al., 2007*). Our data support a hypothesis that higher systemic 5-FU catabolism to due elevated liver DPD expression in carriers of the rs4294451-T allele results in lower exposure to active anti-tumor metabolites of 5-FU. Additionally, we demonstrate that colorectal cancer cell lines likely retain the enhancer functions associated with the rs4294451-T allele, suggesting that tumor cells carrying this variant could more readily inactive 5-FU via increased DPD expression. The higher likelihood of carrying the T-allele in individuals of African ancestry would, therefore, place them at greater risk. Additional studies will be needed to investigate the degree to which rs4294451-T contributes to survival and progression in colorectal cancer and in other solid cancers frequently treated with 5-FU.

The transcription factor CEBPB is a member of the CCAAT Enhancer Binding Protein family, a group of transcription factors that contain basic leucine zipper (bZIP) domains and is highly expressed in liver (*Akira et al., 1990*; *Jakobsen et al., 2013*). Multiple CEBPB isoforms have been detected, with some acting as transcriptional activators and others as inhibitors. The data presented herein indicate that active isoforms of CEBPB are up-regulating *DPYD* through binding to the E9 enhancer region. Inhibitory isoforms of CEBPB have also been shown to be important for certain physiological processes including tumorigenesis and liver regeneration (*Bégay et al., 2015*; *Li et al., 2008*). Our over-expression studies used the full-length active isoform. Therefore, we cannot rule out a role for the inactive isoform participating in the regulation of *DPYD* expression. In addition to acting as a homodimer, CEBPB can also heterodimerize with other CEBP family proteins and interact with other transcription factors, including P300/CBP, CREB, NFKB, AP1, and NFAT, to co-regulate gene expression (*Miller, 2016*; *Seo et al., 2021*). Additional studies are underway to characterize the role of additional regulatory factors within the *DPYD* enhancer region identified in this manuscript.

In silico analyses suggested that the rs4294451 A allele created a stronger binding site for CEBPB within the E9 enhancer region. However, our data demonstrate that the T allele is associated with higher reporter activity (*Figure 3B*), higher DPD expression in both knock-in cells (*Figure 3C*) and human liver tissues (*Figure 3F–G*), and localized epigenetic activation (*Figure 3D–E*). Enrichment of CEBPB to the E9 region was likewise shown to be higher in cells with the rs4294451 T allele (*Figure 5*) and higher levels of interaction were also noted between the E9 region with the T allele and the *DPYD* promoter (*Figure 4*). These findings are also consistent with previously reported eQTL results for the haplotype block linked to rs4294451, where the T allele was associated with higher levels of *DPYD* expression in human liver specimens (*Etheridge et al., 2020*), and our observation that the T allele is associated with cellular resistance to 5-FU (*Figure 4E*). The binding of CEBPB to specific motifs has recently been shown to be cell-type specific and to rely on non-consensus binding motifs with binding strengths that can be modulated by the sequence and structure of surrounding DNA regions (*Cohen et al., 2018*; *Lountos et al., 2022*), providing a possible explanation for the discrepancy between predicted and observed results.

To our knowledge, this is the first report directly linking CEBPB to 5-FU metabolism and the first mechanistic data demonstrating the *cis* effects of a regulatory variant on *DPYD* expression. The role of CEBPB in modulating 5-FU resistance and toxicity is not without precedent. For example, mir-191 is abnormally expressed in several cancers and has been associated with both 5-FU resistance and the regulation of CEBPB expression *Zhang et al., 2015*; however, the CEBPB–*DPYD* regulatory axis has not previously been recognized. Furthermore, CEBPB signaling has been shown to be activated in colorectal cancer cells following treatment with 5-FU (*Wang et al., 2019*), suggesting that CEBPB-mediated activation of *DPYD* expression might represent a dynamic response to therapy.

While the contributions of regulatory variants to 5-FU metabolism have not been widely studied to date, previous studies have explored DPD regulation. Our laboratories previously characterized a *trans*-acting regulatory variant for DPD located within the microRNA mir-27a that was subsequently shown to further increase 5-FU toxicity risk in individuals that carried deleterious nonsynonymous *DPYD* variants (*Offer et al., 2014a*; *Amstutz et al., 2015*). The variant at rs4294451 is in LD with the *DPYD* variants c.85T>C and c.496G>A, which served as coding region proxies for allele-specific expression in our present study (*Figure 3F–G*). The haplotypes defined by these two coding-region variants together with a third variant (c.1129–5923C>G/rs75017182) previously associated with varied levels of systemic DPD activity (*Hamzic et al., 2021*). A subsequent retrospective analysis indicated that these haplotype differences translate to a differential risk of severe 5-FU toxicity (*Medwid et al., 2023*); however, additional studies are needed that are powered to evaluate more than the most common haplotypes. Taken together, these results suggest that the differential regulatory effects of rs4294451 alleles could further impact overall DPD activity and, by extension, modulate the risk of severe 5-FU–related toxicity conferred by coding or splice-variants in *DPYD*. Further work is also needed to identify interactions between the linked variants that impact DPD enzyme activity and the regulatory variant to define systemic DPD function and 5-FU toxicity risk.

## Methods

### Cells

HEK293T, HCT116, and HepG2 cell lines were obtained from the American Tissue Collection Center (ATCC, Manassas, VA). Aliquots of low passage cells were cryopreserved within 2 weeks of receipt. Cells were cultured for no longer than 10 total passages or 2 months, whichever occurred first. All cell lines were monitored for mycoplasma using Hoechst staining (Sigma-Aldrich, St. Louis, MO) at a minimum of one time per month and found to be negative. Culture health and identity were monitored by microscopy to evaluate morphology and by comparing population doubling times to baseline values that were recorded at time of receipt. Additional authentication of cell lines above that described was not performed.

All cell lines were cultured in Dulbecco's modified Eagle's medium (DMEM; Corning, Corning, NY) supplemented with 10% fetal bovine serum (FBS; MilliporeSigma, Billerica, MA), 2 mM L-glutamate (Corning), and 1 x penicillin/streptomycin solution (MilliporeSigma). Cells were maintained at 37 °C with 5% $CO_2$. To support cell attachment and expansion, HepG2 cells were grown on plates coated with 5% Matrigel (Corning).

### Liver tissues

Human liver tissues used for ChIP analyses were processed through Dr. Mary Relling's laboratory at St. Jude Children's Research Hospital, part of the Pharmacogenetics of Anticancer Agents Research (PAAR) Group, and were provided by the Liver Tissue Cell Distribution System funded by NIH Contract #N01-DK-7–0004/HHSN267200700004C and by the Cooperative Human Tissue Network. The acquisition and use of specimens for this manuscript was conducted with the approval of the University of North Carolina at Chapel Hill IRB (study number 10–2253), which has designated the use of these livers for the current analyses as nonhuman subject research and the need for direct consent for use in this study was waived. Human liver tissues used for allele-specific expression were obtained from the University Clinic of Visceral Surgery and Medicine, Inselspital, Bern, Switzerland. Specimens were from donated material from patients who had undergone liver surgery at Inselspital and signed a written consent form for remnant tissues to be used in research (KEK-BE:2016–02202). Patients with impaired liver function due to cirrhosis or other conditions were excluded from analyses.

### Quantitative RT-PCR (RT-qPCR)

Total RNA was extracted from cells using the Direct-zol RNA Kit (Zymo Research, Tustin, CA) following the manufacturer's protocol. Reverse transcription into cDNA was performed using the Transcriptor Reverse Transcriptase kit (Roche, Indianapolis, IN) and random hexamer primers (Roche) according to the manufacturer's instructions. Quantitative PCR (qPCR) was performed on a LightCycler 480 System (Roche) using LightCycler 480 SYBR Green I Master Mix reagents (Roche). Primers used for RT-qPCR are listed in *Supplementary file 1*. RNA expression was normalized to the reference gene L32, and

relative gene expression was calculated using the $2^{-\Delta\Delta CT}$ method. For all analyses, three independent experiments were performed; for each experiment, gene expression was assessed in triplicate.

## Western blot analysis

Whole-cell lysates were extracted using the RIPA lysis buffer system (Santa Cruz Biotechnology, Dallas, TX), separated by SDS-PAGE, and transferred to PVDF membrane (MilliporeSigma). Blots were blocked using Odyssey Blocking Buffer (LI-COR Biosciences, Lincoln, NE) and incubated with primary antibody at 4 °C overnight. Primary antibodies consisted of anti-CEBPB (PA5-27244; 1:1,000 dilution; Thermo Fisher Scientific, Waltham, MA), anti-alpha Tubulin (ab4074; 1:7,500 dilution; Abcam, Waltham, MA), and anti-cas9 (sc-517386; 1:1,000 dilution; Santa Cruz Biotechnology, Dallas, TX). Membranes were washed and incubated with anti-mouse and anti-rabbit secondary antibodies (#926–32212 and #926–68073; both 1:5,000 dilution; LI-COR Biosciences) for 1 hr at room temperature. Blots were imaged using the Odyssey infrared imaging system (LI-COR Biosciences).

## Plasmids

The gRNA expression empty vector lentiGuide-Puro was a gift from Feng Zhang (Addgene_52963). pCMV-FLAG LAP2 was a gift from Joan Massague (Addgene_15738). The oligonucleotides targeting enhancer regions (E9, E16, and E20) were designed using GuideScan (*Perez et al., 2017*), hybridized, phosphorylated, and cloned into lentiGuide-Puro via *Bsm*BI sites. The luciferase expression vector pGL4.10 was purchased from Promega (Madison, WI). Lentivirus vectors expressing shRNAs targeting CEBPB were obtained from the University of Minnesota Genomics Center (Sh1: TRCN0000007440 and Sh2: TRCN0000007442). Plasmids generated for these studies are available from the authors upon request.

## CRISPR inactivation (CRISPRi) and activation (CRISPRa)

HepG2 cell lines that overexpress dCas9-KRAB and dCas9-P300 were generated by lentiviral transduction using lenti-EF1a-dCas9-KRAB-Puro and pLV-dCas9-p300-P2A-PuroR, respectively. To generate lentiviral particles, HEK293T cells were co-transfected with lenti-EF1a-dCas9-KRAB-Puro plasmid (a gift from Kristen Brennand; Addgene_99372) or pLV-dCas9-p300-P2A-PuroR plasmid (a gift from Charles Gersbach; Addgene_83889), psPAX2 (a gift from Didier Trono; Addgene_12260), and pMD2.G (a gift from Didier Trono; Addgene_12259) using TransIT-Lenti Transfection Reagent (Mirus Bio, Madison, WI). A 3:1 ratio of transfection reagent to total plasmid was used for all transfections. For all transfections, medium was changed 14 hr after transfection, and viral supernatants were collected 34 hr later. Supernatants were filtered using 0.45 µm PVDF filters (MilliporeSigma) to remove debris/cells and used directly for transductions. For transductions, HCT116 cells or HepG2 cells were seeded at a density of $4 \times 10^5$ cells per well in 6-well plates and incubated with 500 µL virus-containing supernatant, 12.5 µg/mL polybrene (MilliporeSigma), and 1.5 mL fresh DMEM culture medium. Medium was changed after 24 hr. Cells were treated with 1 µg/mL puromycin to initiate selection for transduced cells 48 hr after transduction. Expression of dCas9-KRAB and dCas9-P300 was confirmed by western blotting. Guide RNAs (gRNAs) for each target region were identified and designed using GuideScan (*Perez et al., 2017*). Oligonucleotides corresponding to each gRNA were obtained from IDT (Coralville, IA), hybridized, phosphorylated using T4 polynucleotide kinase (New England Biolabs, Ipswich, MA), and ligated into digested BsmBI-digested (New England Biolabs) lentiGuide-Puro vector (a gift from Feng Zhang; Addgene_52963). Cell lines stably expressing dCas9-KRAB or dCas9-P300 were transfected with plasmids encoding gRNAs or lentiGuide empty vector using TransIT-X2 (Mirus Bio). RNA was extracted 2 days after transfection, and *DPYD* expression was measured by RT-qPCR.

## Chromatin immunoprecipitation coupled with quantitative PCR (ChIP-qPCR)

ChIP assays were performed using the ChIP-IT Express Enzymatic Kit (Active Motif, Carlsbad, CA) following the manufacturer's directions. One million cells were harvested, washed, and cross-linked using 1% formaldehyde (Thermo Fisher Scientific) in serum-free medium for 10 min followed by quenching with 125 mM glycine for 5 min at room temperature. The chromatin was digested using an enzymatic shearing cocktail provided by the kit to an average size of 200–1000 bp. Two percent

of the sheared chromatin was retained as input control. Approximately 25 μg sheared chromatin was incubated with 2 μg H3K27ac antibody (ab4729; Abcam), 2 μg H3K9me3 antibody (ab8898; Abcam), 2 μg CEBPB antibody (PA5-27244; Thermo Fisher Scientific), or 2 μg control normal Rabbit IgG antibody (antibody 2729; Cell Signaling Technology, Danvers, MA), in the presence of protein G magnetic beads, ChIP buffer, and protease inhibitor cocktail (all Active Motif) for 4 hr at 4 °C. Magnetic beads were washed, chromatin was eluted, cross-linking was reversed, and proteinase K treatment was performed using reagents provided in the kit following the manufacturer's directions. DNA was purified using the QIAquick PCR Purification Kit (Qiagen, Germantown, MD). Purified DNA was used for subsequent qPCR reactions using SYBR Green I Master Mix on a LightCycler 480 System (Roche). Enrichment was calculated using the following formula: (a) % ChIP = $2^{(\text{Input Ct - ChIP Ct})}$ * (dilution factor) (100); (b) % IgG = $2^{(\text{Input Ct - IgG Ct})}$ * (dilution factor) (100); (c) Fold Enrichment = % ChIP ÷ % IgG. Primers used for ChIP-qPCR are listed in *Supplementary file 1*.

## Luciferase reporter assays

The *DPYD* promoter region, consisting of the 1154 bp of genomic DNA directly upstream of the *DPYD* TSS, was amplified by PCR, digested with EcoRV and HindIII (New England Biolabs), and cloned into compatible sites on the pGL4.10 vector (Promega). The 1392 bp region comprising the E9 region was PCR amplified from genomic DNA and cloned upstream of the *DPYD* promoter using KpnI and SacI sites (New England Biolabs). A vector containing the A allele of rs4294451 within E9 was confirmed by Sanger sequencing. The vector containing the rs4294451 T allele was generated by site-directed mutagenesis and confirmed by sequencing. All primers used in vector construction are listed in *Supplementary file 1*. For reporter assays, $10^5$ HEK293T cells were seeded into 24-well plates and co-transfected with pGL4.10-based plasmids and pRL-SV40 Renilla luciferase plasmid (Promega). After 48 hr, luciferase activity was measured using the Dual-Glo Luciferase Assay (Promega) following the manufacturer's recommendations on a Synergy HTX Multimode Plate Reader (Agilent Technologies, Santa Clara, CA).

## Knock-in cell lines for rs4294451 genotypes

Knock-in cell lines were generated using CRISPR/Cas9 gene editing. Homology-directed repair (HDR) donor templates and target-specific Alt-R crRNA were designed using the Alt-R HDR Design Tool (IDT). Equimolar amounts of crRNA (IDT) and common Alt-R tracrRNA (IDT) were annealed to form the gRNA duplex. RNP complexes were formed by combining gRNA with Alt-R S.p. Cas9 Nuclease V3 (IDT) to a final Cas9:gRNA ratio of 4:4.8. RNA complex and Alt-R HDR Donor Oligos were transfected into HCT116 cells by electroporation (Lonza Nucleofector 96-well Shuttle System; Lonza, Bend, OR) using parameters provided by the manufacturer. Seventy-two hours after transfection, serial dilutions were performed to obtain single-cell clones. Clones were expanded, genomic DNA was isolated, and rs42944551 genotype was determined by rhAmp Genotyping (assay ID: Hs.GT.rs4294451.A.1; IDT) using rhAmp Genotyping Master Mix and universal probe Reporter Mix (both IDT). Cell morphology was compared to the parental cell lines and confirmed to be consistent.

Upon derivation of confirmed knock-in clones, low passage cells were cryopreserved. Cell morphology was compared to that of parental cultures and found to be consistent. Consistent population doubling times were also noted. Cells were cultured for no longer than 10 total passages following the establishment of a confirmed clone. All cell lines were monitored for mycoplasma using Hoechst staining (Sigma-Aldrich) at a minimum of one time per month and found to be negative. Additional authentication of cell lines above that described was not performed.

## Allele-specific gene expression

Allele-specific expression of *DPYD* was measured using reverse transcriptase droplet digital PCR (RT-ddPCR) by targeting variants in the coding region of *DPYD* (c.85T>C and c.496A>G). Linkage disequilibrium (LD) between variants was calculated using LDpair implemented within LDlink (*Machiela and Chanock, 2015*). DNA was extracted from donor liver tissues using the QIAamp DNA Mini Kit (Qiagen). For RNA extraction, tissues were lysed in QIAzol (Qiagen), and RNA was extracted using the miRneasy Kit (Qiagen) with on-column DNA digestion using RNase-free DNase (Qiagen). RNA quality was assessed using an Agilent 2100 Bioanalyzer running 2100 Expert Software v.B.02.10 using Agilent RNA 6000 Nano kits.

DNA samples were genotyped for c.85, c.496, and rs4294451 loci using TaqMan SNP Genotyping assays (Applied Biosystems, Waltham, MA). Tissues that were heterozygous for at least one of the coding region SNPs (i.e. c.85T>C and/or c.496A>G) were suitable for allele-specific expression analysis because expression from both alleles could be discriminated using the coding-region markers in mRNA. Allele-specific expression was measured using RT-ddPCR with the One-Step RT-ddPCR Advanced Kit for Probes (Qiagen) on a QX200 ddPCR Droplet Reader (BioRad, Hercules, CA). Fractional abundance was calculated using QuantaSoft software (BioRad). Poisson distributions were determined using Quantasoft and were used to define 95% confidence intervals. To address possible biases associated with differing probe efficacies caused by differential probe binding affinities or specificities, fractional abundances are also reported relative to those measured in DNA.

## Quantitative analysis of chromatin conformation capture (3C-qPCR)

Quantitative analysis of chromatin conformation capture (3C-qPCR) was performed as described by *Hagège et al., 2007* with minor modifications. Cells were trypsinized using 0.25% w/v trypsin-EDTA (Thermo Fisher Scientific) and resuspended in 1% FBS (MilliporeSigma) in DMEM (Corning) for counting by flow cytometry using a NovoCyte 3000 RYB system (Agilent Technologies). Cells ($10^7$) were pelleted by centrifugation at 300×$g$ at 22 °C for 5 min. Supernatant was discarded, and cell pellets were resuspended in 500 µL of 10% FBS in DMEM. Single-cell suspensions were obtained by filtration through a 40 µm cell strainer (Corning). Crosslinking was performed by adding 9.5 mL of 1% formaldehyde in 10% FBS in PBS per $10^7$ cells. Reactions were incubated at 22 °C while tumbling for 10 min. Reactions were transferred to ice and crosslinking was quenched by the addition of ice-cold glycine (MilliporeSigma) to achieve a final concentration of 0.125 M. Samples were centrifuged at 300×$g$ at 4 °C for 5 min. Supernatant was removed and discarded. Crosslinked cell pellets were lysed in 5 mL cold lysis buffer (10 mM Tris-HCl, pH7.5; 10 mM NaCl; 5 mM MgCl; 0.1 mM EGTA; 1 x Roche Complete protease inhibitor cocktail). Lysis was allowed to proceed for 10 min on ice with intermittent gentle pipetting to obtain homogeneous suspensions of nuclei. Nuclei were pelleted at 500×$g$ at 4 °C for 5 min and resuspended in 0.5 mL of 1.2 x NEBuffer r2.1 (New England Biolabs) containing 3% sodium dodecyl sulfate (SDS, MilliporeSigma). Samples were incubated at 37 °C for 1 hr with shaking. Following incubation, Triton X-100 (MilliporeSigma) was added to a final concentration of 2%. Reactions were incubated at 37 °C for 1 hr with shaking. To digest DNA, 600 U of HindIII (New England Biolabs) was added, and reactions were incubated for 16 hr at 37 °C with shaking. To inactivate digestion, SDS was added to a final concentration of 1.6%, and reactions were incubated at 65 °C for 25 min with shaking. Excess SDS was sequestered by the addition of Triton X-100 to a final concentration of 1%. Samples were divided into two aliquots, one for ligation and the other for non-ligation control. Ligation was performed on a sevenfold dilution of HindIII-digested chromatin using 100 units of Quick T4 DNA ligase (New England Biolabs) at 16 °C for 16 hr, followed by 1 hr at 22 °C. Proteinase K (300 µg; New England Biolabs) was added to ligation mixtures and non-ligated controls, and samples were incubated at 65 °C for 16 hr to reverse crosslinking. DNA was subsequently purified by adding 300 µg RNase A (Thermo Fisher) followed by a 30 min incubation at 37 °C and subsequent phenol-chloroform extraction as described (*Hagège et al., 2007*). DNA pellets were washed with 70% ethanol and dissolved in 10 mM Tris pH 7.5.

For qPCR, directional primers were designed within each fragment as depicted in *Figure 4A*. Two anchors, one localized to the fragment containing the *DPYD* promoter and the other to the E9 region, were selected and paired with primers designed to 'walk' across the length of the surrounding region. Reactions were carried out in 10 µL reaction volumes, consisting of 5 pmol of each primer and 1 µL of a 1:50 dilution of each 3 C sample. Amplification was performed using LightCycler 480 SYBR Green I Master Mix (Roche) on a LightCycler 480 thermocycler (Roche). PCR conditions were 95 °C for 10 min, and 40 cycles of 95 °C for 15 s, 65 °C for 1 min, and 72 °C for 15 s. Enrichment was calculated as $2^{-(cp\ (ligated\ DNA)\ -\ cp\ (non-ligated\ DNA))}$. Enrichment with the fragment containing the nearest primer was used as the control for interaction frequency for further normalization between replicate experiments. Specifically, when primer 5 was used as the anchor, data were normalized to the average of values obtained from using primers 5 and 4. When primer 7 was used as the anchor, data were normalized to the average of values from primers 7 and 8. All experiments were performed in triplicate. Primer sequences and positions relative to the *DPYD* transcription start site (TSS) for qPCR are listed in *Supplementary file 2*.

## Cellular sensitivity to 5-FU

Cell viability was monitored using the xCELLigence MP Real-Time Cell Analysis (RTCA) system (Agilent) as previously described (*Shrestha et al., 2018*). Background impedance values for RTCA E-View plates (Agilent) were obtained using complete DMEM prior to plating cells. Cells were seeded at a density of 5000 cells per well and incubated for 20 hr, at which time medium was removed and replaced with medium containing serial dilutions of 5-FU ranging from 1.25 µM to 100 µM. Impedance values (expressed as cell index, CI, units) were recorded every 15 min over the course of the experiment to monitor proliferation. Results represent the average of three independent cultures. To account for minute differences in plating and potential cell loss during drug addition, relative CI was calculated as the CI measured 48 hr after 5-FU divided by the CI recorded immediately after treatment. Four parameter logistic non-linear regression analysis was used to determine $IC_{50}$ concentrations (GraphPad Prism version 9, GraphPad Software, San Diego, CA).

## CEBPB knockdown cells

Lentiviral particles for CEBPB knockdown were generated by transfecting shRNA plasmids (TRCN0000007440 and TRCN0000007442) or scramble shRNA control (a gift from David Sabatini; Addgene_1864 *Sarbassov et al., 2005*), psPAX2 (a gift from Didier Trono; Addgene_12260), and pMD2.G (a gift from Didier Trono; Addgene_12259) into HEK293T cells using TransIT-Lenti (Mirus Bio, Madison, WI). A 3:1 ratio of transfection reagent to total plasmid was used for all transfections. For all transfections, medium was changed 14 hr after transfection, and viral supernatants were collected 34 hr later. Supernatants were filtered using 0.45 µm PVDF filters (Millipore) to remove debris/cells and used directly for transductions. For transductions, target cells were seeded at a density of $4 \times 10^5$ cells per well in six-well plates and incubated with 500 µL virus-containing supernatant, 12.5 µg/mL polybrene (MilliporeSigma), and 1.5 mL fresh DMEM culture medium.

## Analysis of transcription factor binding motifs

Potential transcription factor binding sites within the E9 region were determined using TFBSTools (*Tan and Lenhard, 2016*). DNA sequence corresponding to the 101-nucleotide region centered on rs4294451 was retrieved from genome build GRCh38.p13 (NC_000001.11). A second DNA string was created to mimic the sequence corresponding to the rs4294451 A allele. Position frequency matrices for each transcription factor were retrieved from JASPAR CORE 2022 (*Castro-Mondragon et al., 2022*) and converted to log-scale position weight matrices using the toPWM method implemented in TFBSTools. JASPAR includes binding site information from multiple sources. ENCODE data (*ENCODE Project Consortium, 2012*) were available for CEBPB (matrix ID MA04661) and JUND (MA0491.1). In the absence of ENCODE data, alternatives including PAZAR (*Portales-Casamar et al., 2007*) (HNF4A, MA0114.2; FOXA1, MA0148.3), REMAP (*Hammal et al., 2022*) (FOXA2, MA0047.3), and data from an individual publication (*Jolma et al., 2013*) (MYBL2, MA0777.1) were used. EP300 is a cofactor that does not recognize a specific DNA motif on its own; instead, it interacts with various DNA-binding factors to modify chromatin and facilitate the activation of target genes. As such, P300 does not have a DNA-binding motif and was not included in analysis. Nucleotide sequences were scanned using the patterns presented in the position weight matrices to identify putative transcription factor binding sites. Forward and reverse strands were searched, and the 70th percentile between the minimum and maximum possible value for a matrix was used as the minimum threshold score. Empirical p-values for each score were calculated by an exact method using the TFMPvalue R package. R version 4.2.2 was used for analyses.

## Statistical analyses

Significance was defined as $P < 0.05$ unless otherwise noted in the text. Pairwise comparisons were performed using unpaired two-tailed Student's t-tests calculated using GraphPad Prism version 9. One-way ANOVA statistics were calculated using GraphPad Prism. Summary statistics pertaining to allele-specific expression were calculated using Quantasoft software as described in the Allele-specific gene expression section. Transcription factor binding predictions and associated analyses were performed in R version 4.2.2 as described above.

# Acknowledgements

This work was supported by the National Cancer Institute of the National Institutes of Health under award number R01CA251065 (SMO, PI). Allele-specific expression experiments were supported by the Swiss National Science Foundation under award number 320030_212583 (CRL, PI). The funding bodies did not contribute to the design of the study, the collection, analysis, and interpretation of data, or in writing the manuscript.

## Additional information

### Competing interests

Federico Innocenti: Currently an AbbVie employee and receives stocks from the company. Steven M Offer: Reports current research support from Processa Pharmaceuticals, Inc, which are outside of the conduct of this study. Has previously served as an independent consultant for Processa Pharmaceuticals, Inc, in matters not related to this study. The other authors declare that no competing interests exist.

### Funding

| Funder | Grant reference number | Author |
|---|---|---|
| National Cancer Institute | R01CA251065 | Steven M Offer |
| Schweizerischer Nationalfonds zur Förderung der Wissenschaftlichen Forschung | 320030_212583 | Carlo R Largiadèr |

The funders had no role in study design, data collection and interpretation, or the decision to submit the work for publication.

### Author contributions

Ting Zhang, Conceptualization, Formal analysis, Supervision, Investigation, Visualization, Methodology, Writing – original draft, Project administration, Writing – review and editing; Alisa Ambrodji, Data curation, Formal analysis, Investigation, Methodology, Writing – original draft; Huixing Huang, Zoey B Temesgen, Data curation, Investigation; Kelly J Bouchonville, Brianna M Bembenek, Data curation, Formal analysis, Investigation, Methodology, Writing – review and editing; Amy S Etheridge, Resources, Data curation, Writing – original draft, Writing – review and editing; Remington E Schmidt, Data curation, Formal analysis, Investigation, Methodology; Zhiquan Wang, Data curation, Investigation, Methodology; Federico Innocenti, Resources, Supervision; Deborah Stroka, Resources, Methodology; Robert B Diasio, Project administration; Carlo R Largiadèr, Conceptualization, Resources, Formal analysis, Supervision, Funding acquisition, Investigation, Visualization, Methodology, Writing – original draft, Project administration, Writing – review and editing; Steven M Offer, Conceptualization, Data curation, Formal analysis, Supervision, Funding acquisition, Investigation, Visualization, Methodology, Writing – original draft, Project administration, Writing – review and editing

### Author ORCIDs

Remington E Schmidt ⓘ http://orcid.org/0000-0001-6933-4226
Brianna M Bembenek ⓘ http://orcid.org/0000-0002-8798-8706
Deborah Stroka ⓘ http://orcid.org/0000-0002-3517-3871
Carlo R Largiadèr ⓘ http://orcid.org/0000-0002-0889-8922
Steven M Offer ⓘ https://orcid.org/0000-0002-7513-9678

Joint Public Review: https://doi.org/10.7554/eLife.94075.3.sa1
Author response https://doi.org/10.7554/eLife.94075.3.sa2

## Additional files

### Supplementary files
• Supplementary file 1. Table of primers used for chromatin immunoprecipitation (ChIP), cloning, quantitative PCR (qPCR), and site-directed mutagenesis.
• Supplementary file 2. Table of sequences and positions of the primers used for 3 C analysis.
• MDAR checklist

### Data availability
All data generated or analysed during this study are included in the manuscript and supporting files. All uses of previously published data within this manuscript are noted in the text and/or figure legends. Data were retrieved from GeneHancer (http://www.genecards.org/, https://genecards.weizmann.ac.il/geneloc/index.shtml), the Ensembl Regulatory Build (https://grch37.ensembl.org/info/genome/funcgen/regulatory_build.html) and the Genome Aggregation Database (gnomAD v3.1.2 and gnomAD v3.1, https://gnomad.broadinstitute.org/).

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
