## [Editor Report · eLife assessment]

This manuscript presents **valuable** findings on the identification of epigenetically mediated control for the recognition of dihydropyrimidine dehydrogenase (DPYD) gene expression that is linked with cancer treatment resistance using 5-fluorouracil. The evidence is **compelling**, supported by data from patient-derived specimens and direct assessment of 5-fluorouracil sensitivity, which provides confidence in the proposed mechanisms. The model is additionally supported by genome data from a population with high "compromised allele frequency". This work will interest those studying drug resistance in cancer therapy.

---

## [Referee Report · Joint Public Review]

Zhang et. al. presents compelling results that support the identification of epigenetically mediated control for the recognition of dihydropyrimidine dehydrogenase (DPYD) gene expression that is linked with cancer treatment resistance 5-fluorouracil. The experimental approach was developed and pursued with in vitro and in vivo strategies. Combining molecular, cellular, and biochemical approaches, the authors identify a germline variant with compromised enhancer control. Several lines of evidence were presented that are consistent with increased CEBP recruitment to the DPYD regulatory domain with consequential modifications in promoter-enhancer interactions that are associated with compromised 5-fluorouracil resistance. Functional identification of promoter and enhancer elements was validated by CRISPRi and CRISPRa assays. ChIP and qPCR documented histone marks that can account for the control of DPYD gene expression were established. Consistency with data from patient-derived specimens and direct assessment of 5-fluorouracil sensitivity provides confidence in the proposed mechanisms. The model is additionally supported by genome data from a population with high "compromised allele frequency". It can be informative to directly demonstrate DPYD promoter-enhancer interactions. However, the genetic variants support the integration of regulatory activities.

---

## [Author Response]

The following is the authors’ response to the previous reviews.

**Reviewer #1 (Recommendations For The Authors):**
The data is poorly dealt with, and the figures are shown poorly. For example, Figure 2A is not even shown totally.

We apologize for any difficulties that the reviewer encountered while attempting to view the figures. We have confirmed that all figures, including all panels of Figure 2, display correctly on the HTML and PDF versions of the article hosted at bioRxiv. The HTML and PDF versions generated by eLife also appears to contain all figures and panels in their entirety.

**Reviewer #2 (Recommendations For The Authors):**
Please refer to the public review for possible revisions.

We thank Reviewer #2 for the summary and thoughtful comments provided in the Public Review. We note the point of possible revision noted from the Public Review: “It can be informative to directly demonstrate DPYD promoter-enhancer interactions. However, the genetic variants support the integration of regulatory activities.” In Figure 4, we provide evidence for direct promoterenhancer interaction though the use of 3C. We furthermore demonstrate that these interactions are dependent upon genotype at rs4294451 as stated by the reviewer. We have highlighted the promoter-enhancer interaction in the revised manuscript, lines 323-325. The role of genotype in this interaction is also specifically discussed in lines 378-381.